# BDNF Modulation by microRNAs: An Update on the *Experimental* Evidence

**DOI:** 10.3390/cells13100880

**Published:** 2024-05-20

**Authors:** Gilmara Gomes De Assis, Eugenia Murawska-Ciałowicz

**Affiliations:** 1Department of Restorative Dentistry, Araraquara School of Dentistry, São Paulo State University (UNESP), Araraquara 14801-385, SP, Brazil; 2Department of Physiology and Biochemistry, Wroclaw University of Health and Sport Sciences, 51-612 Wrocław, Poland; eugenia.murawska-cialowicz@awf.wroc.pl

**Keywords:** BDNF, microRNA, regenerative medicine, neuroprotection

## Abstract

MicroRNAs can interfere with protein function by suppressing their messenger RNA translation or the synthesis of its related factors. The function of brain-derived neurotrophic factor (BDNF) is essential to the proper formation and function of the nervous system and is seen to be regulated by many microRNAs. However, understanding how microRNAs influence BDNF actions within cells requires a wider comprehension of their integrative regulatory mechanisms. **Aim**: In this literature review, we have synthesized the evidence of microRNA regulation on BDNF in cells and tissues, and provided an analytical discussion about direct and indirect mechanisms that appeared to be involved in BDNF regulation by microRNAs. **Methods**: Searches were conducted on PubMed.gov using the terms “BDNF” AND “MicroRNA” and “brain-derived neurotrophic factor” AND “MicroRNA”, updated on 1 September 2023. Papers without open access were requested from the authors. One hundred and seventy-one papers were included for review and discussion. Results and **Discussion**: The local regulation of BDNF by microRNAs involves a complex interaction between a series of microRNAs with target proteins that can either inhibit or enhance BDNF expression, at the core of cell metabolism. Therefore, understanding this homeostatic balance provides resources for the future development of vector-delivery-based therapies for the neuroprotective effects of BDNF.

## 1. Introduction

MicroRNAs are a class of non-coding RNAs which do not code for proteins but carry out their biological function by regulating the cell proteome at the translational level. They can be expressed within the activation of a gene promoter or their own promoters [1,2], and play a regulatory role in protein synthesis by targeting and degrading RNA transcripts containing compatible nucleotide sequences [3,4,5,6,7]. MicroRNAs are seen to participate in the functional regulation at the distant synaptic sites in neuronal cells, and to modulate inflammatory mechanisms that lead to neurological diseases [8,9].

As a main expressed neurotrophin, brain-derived neurotrophic factor (BDNF) plays essential roles in the development and maintenance of neural tissues [10,11]. The signaling of the mature form of BDNF via tropomyosin receptor kinase (Trk) B participates in neuronal survival, dendritogenesis, synaptogenesis, axon growth, and synaptic function; meanwhile, the release of a pro-BDNF isoform can bind with a low affinity to p75 neurotrophin receptor (p75NTR) and lead to apoptosis, so that a tight regulation of BDNF activity is necessary for the proper functioning of the central nervous system (CNS) [12,13,14].

Analyses in silico estimate hundreds of microRNAs as possible regulators of BDNF. However, with a 10–20% variability detected in the predicted regulatory relationships between genes and microRNAs in the human RefSeq dataset, the effective regulation of BDNF mRNA transcripts by microRNAs in biological systems is much smaller [15]. In addition, the microRNA affinity for multiple targets and microRNA–microRNA interactions in a cell milieu influence their regulation and cannot be predicted by computational logarithms. As the studies typically address only one or a few microRNAs in their experiments, it remains a challenge to design pre-clinical studies based on computational predictions [2,16].

Regarding the notion that microRNAs target and degrade RNA transcripts across different tissues and regulate protein function via direct and indirect mechanisms; a broader look at the possible scenarios where BDNF actions can potentially be affected by microRNAs shall include all the available data reporting interactions between microRNAs and BDNF in biological experiments. Further, understanding how microRNAs effectively regulate BDNF actions provides the basis for the development of potential therapies against neurodegenerative conditions. Therefore, we have collected all the available data on the post-transcriptional regulation of BDNF by microRNAs evidenced in experimental studies, and provided a synthesis of the regulatory mechanisms currently demonstrated.

## 2. Methods

In order to retire all the scientific publications possibly reporting data from the analysis of microRNAs and BDNF expression in a same biological system, a systematic search was conducted on PubMed.gov using the following combination of terms: [“BDNF” AND “MicroRNA”] OR [“brain-derived neurotrophic factor” AND “MicroRNA”]. All the available publications were retrieved and screened by abstract. Papers published without open access were requested from the authors via email or ResearchGate. Studies containing data from BDNF and microRNA analyses in vivo or in vitro were considered for inclusion. The studies reporting data of microRNAs that did not influence BDNF regulation, reviews, articles not written in English, high-throughput profiles, and computational predictor studies, as well as those not made available by the authors, were excluded from discussion during full-text assessment. The last search update, performed in 1 September 2023, launched 314 papers published from 2006 to 2023 indexed in PubMed (see Figure 1). The studies selection was performed using the software Mendeley 1.19.8.

Two-hundred and ninety-seven articles were sought for retrieval according to the inclusion criteria of containing data from BDNF and microRNA analyses, after removal of duplicates. Two hundred and sixty-one papers were assessed by full text. A total of 171 were found to include the analyses of BDNF and diverse microRNAs and were included in the qualitative synthesis, after exclusion criteria (Figure 1). A list of the studies and microRNAs involved in BDNF regulation was displayed in Table 1.

## 3. Discussion

### 3.1. BDNF by microRNAs

A great number of microRNAs are able to target BDNF mRNA transcripts or influence BDNF activity. The post-transcriptional regulation of BDNF can influence BDNF synthesis and activity in a non-specific manner throughout tissues. Moreover, in addition to the ability to target and degrade the transcripts of BDNF mRNA in ‘direct regulation’, microRNAs can affect the activity of BDNF (either positively or negatively) via the regulation of other factors, here referred to as ‘indirect regulation’ (Figure 2). A great part of the research identifying BDNF-target microRNAs concerns investigations on the oncogenicity of BDNF/TrkB signal transduction in tumor cell growth and metastasis [188,189]. A list of microRNAs found to degrade BDNF mRNA transcripts in oncology research is as follows: miR-10a, miR-22, miR-204, miR-107, miR-382, miR-496, miR-497, miR-584, miR-744, miR-26a-1, and miR-26a-2 subtypes [24,28,58,60,72,75,78,88,93,94,96,102,112,125,126,128,132,133,137,190]. The evidence that miR-206 is able to suppress BDNF synthesis in diverse tissues such as the cardiac muscles, the skeletal muscles, and the endothelial tissue elucidates a role for microRNAs in tissue–tissue communication, although their actions might be locally regulated [30,43,45,48,61,92,102,115,120,139,142,169,171,177,191].

Some microRNAs are released from cells by membrane-derived vesicles, lipoproteins, and other ribonucleoprotein complexes and travel through the blood stream reaching recipient cells in distant tissues [192], providing communication between disparate cell types and diverse biological mechanisms and homeostatic pathways. Our data collection reports a number of microRNAs whose circulating levels are increased in inflammatory conditions and that demonstrably suppress BDNF synthesis in the CNS, namely, miR-1, miR-128, miR-182-5p, miR-195-5p, and miR-451a [55,151,155,174,178]. Among those circulating microRNAs which suppress BDNF synthesis, some were found to be involved in the physiopathology of neuropsychiatric disorders and disease such as anxiety/depression—miR-182, miR-206-3p, miR-1-3p, MiR-16, miR-124, miR-432, and miR-182 [29,36,37,113,160,184,187,188]; schizophrenia—miR-16, miR-195, and miR-30a-5p [19,20,165,193]; Parkinson’s disease—miR-494-3p [144]; and dementia—miR-10a, miR-34a-5p, miR-204, and miR-613 [40,50,84,89,121,123,124,194]. Together, those findings suggest that a mechanistic *crosstalk* between inflammation in peripheral tissues and neurodegenerative conditions in the CNS might be driven by microRNAs.

MicroRNAs play crucial roles in immunoinflammatory reactions. In normal conditions, the CNS parenchyma is not exposed to peripheral immune cells or robust inflammatory responses, and the microglia and astrocytes remain quiescent. However, upon stress, the astrocytes and microglia transiently activate and produce chemokines and cytokines, and other small-molecule messengers (prostaglandins, nitric oxide, and reactive oxygen species—ROS) which contribute to the inflammatory response and subsequent restoration of CNS homeostasis [195]. The study by Kynast and colleagues [38] identified that miR-124 is constitutively expressed in neurons of the dorsal horn in the spinal cord, where its elevation is associated with a decrease in BDNF levels, while a decrease in miR-124 levels leads to the elevation in methyl CpG binding protein 2 (MeCP2) and BDNF expression levels. From a different perspective, the studies by [129,196] demonstrated that miR-124 is able to attenuate an acute increase in pro-inflammatory factors in the CNS by suppressing the early growth response 1 (EGR1) and preventing a decline in BDNF expression. Conversely, Yu et al. [180] showed that BDNF administration increased the expression levels of miR-3168, and suppressed the secretion of interleukin (IL)-1β, TNF-α, and IL-6 in the activated macrophage.

Another mechanism by which microRNAs indirectly modulate BDNF synthesis in inflammatory conditions involve the Let-7 miRNA family [197], which include let-7a, let-7b, let-7c, let-7d, let-7e, let-7f, let-7g, let-7i, miR-98, and miR-202 [198]. The dysregulation of let-7 leads to a less differentiated cellular state and cell-based diseases such as cancer. Cho and colleagues [53] investigation in neural tissue reported that let-7a levels increase in microglia following the accumulation of ROS and pro-inflammatory cytokines. The data indicated that let-7a participates in reducing nitrite production while increasing the levels of inducible NO synthase and IL-6. Anti-inflammatory events accompanied an upregulation in BDNF expression levels. Alternatively, Nguyen and colleagues [111] detected that miR-let-7i suppresses the synthesis of progesterone receptor membrane component 1, reducing the progesterone-inducible release of BDNF by astrocytes. Such a reduction has a negative effect on neuronal tissue recovery. These findings show that Let-7 members might exert specific roles that positively or negatively affect BDNF function in the CNS parenchyma.

Although the regulation of protein function by microRNAs mostly always depends on their nucleotide sequence to target mRNA transcripts present in the same micro environment, in silico predictions of BDNF-target microRNAs are not always confirmed in biological systems. Meanwhile, some experiments have pointed out that the microRNA targeting of BDNF mRNA is selectively guided by their prime untranslated region (3′-UTR) [28,51,127,130]. Having noted the presence of two variants of 3′ UTR regions in the mRNA transcripts of BDNF, which exert an influence on their cellular trafficking/localization [199], the mechanistic regulation of BDNF by microRNAs within a cell might as well occur in a local specificity manner, a least in cells that express the two BDNF mRNA 3′ UTR isoforms.

### 3.2. Neuroplasticity and BDNF Regulation by microRNAs

The expression of BDNF is present in progenitor cells from the early embryonic phase and in neural tissue throughout the whole lifespan. Its participation in essential processes such as dendritogenesis, axonal innervation and synaptogenesis, neuronal growth, and survival guarantees the maintenance and proper functioning of the neuronal tissue [200,201,202].

A growing number of microRNAs have been identified as direct regulators of BDNF in the neural tissue. Here, we list some of the microRNAs that target and degrade BDNF mRNA transcripts and modulate BDNF actions in processes such as neuronal cell growth, differentiation, and proliferation: miR-1, miR-1b, miR-1-3p, miR-10a, miR-10b, miR-10a-5p, miR-15a, miR-16, miR-26a, miR-34a, miR-103, miR-125b, miR-125b-5p, miR-30a-5p, miR-34a-5p, miR-140, miR-139-5p, miR-155, miR-191, miR-186, miR-191, miR-191-5p, miR-191a-5p, miR-204-5p, miR-206, miR-210, miR-210-3p miR-211, miR-216a-5p, miR-219, miR-636-3p, miR-365, miR-375, miR-551b-5p, miR-937, miR-497, and the miR-497a subtypes [27,39,41,52,57,63,68,71,73,76,79,86,87,93,95,99,107,109,116,120,125,131,134,135,140,141,145,146,147,149,150,152,153,154,156,158,159,161,163,166,167,168,173,175,176,179,183,186,203,204,205,206,207,208,209,210,211].

Some studies’ evidence shows that Sonic hedgehog (Shh), a basic protein expressed in the mid-line CNS as an inductive signal in the patterning of the ventral neural tube, the anterior–posterior limb axis, and the ventral somites [212], is able to relieve the suppression of miR-206 on BDNF mRNA translation, which, in turn, enhances BDNF-TrkB signaling during the differentiation and innervation processes, as seen in muscle cells [26,34]. Shh is a key signaling molecule in the embryonic morphogenesis and organization of the nervous system. Its signaling via the receptor patched-mediated–smoothened receptor complex is putative to the development of the neural tube, while the abnormal activation of Shh signaling implicates various types of cancers [212]. This indirect and positive effect of Shh on BDNF activity seems to be involved in a complex and phasic destabilization of cell homeostasis during differentiation in mesenchymal cells.

BDNF binding to TrkB receptors at the neuronal cells’ surface leads to the dimerization and transphosphorylation of a critical regulator of actin dynamics in the axons and dendrites named LIM domain kinase 1 (LIMK1). This occurs independently of TrkB kinase activity. The LIMK1 mRNA transcript is a target for miR-134 in the axon and dendrite cell compartments, and is able to annul BDNF/TrkB-induced protein synthesis during the synaptic activity, whenever TrkB activation is not sufficient to surpass miR-134 suppression on LIMK1. This suggests that miR-134 actively participates in competitive synapses formation, and establishes a role for this microRNA in the fine-tuned regulation of neuroplasticity processes [17,23,25,62,80,213,214].

Several microRNAs were found to target different components of BDNF-TrkB signaling intracellular cascades, consequently decreasing the activity of the cAMP response element binding (CREB) protein, leading to a decrease in *BDNF* gene expression. The investigation by Thomas et al. [90] identified that miR-137 regulates the levels of various proteins within the PI3K-Akt-mTOR pathway in neurons, namely, p55g, PTEN, Akt2, GSK3b, mTOR, and rictor. And this negatively affects the BDNF-induced dendritic outgrowth. In addition, miR-221, miR-383, and miR-199a-5p were shown to suppress the synthesis of Wnt2, which is a glycoprotein with essential roles in embryonic development and dendrite development [215]. The neuronal activity enhances the CREB-dependent transcription of Wnt2, which, in turn, stimulates dendritic arborization. Both Wnt2 and BDNF are CREB-responsive genes, and, thus, Wn2t suppression results in a decrease in BDNF expression possibly via the Wnt2/CREB/BDNF axis [108,216,217]. Additionally, some microRNAs were reported to be negatively correlated with the levels of BDNF in studies, i.e., miR-183/96 [44,54], miR-134 [105,122,138], and miR-182-5p [174,182].

### 3.3. Cell Metabolism and BDNF Regulation by microRNAs

The post-transcriptional regulation of proteins elicits compensatory mechanisms to maintain the transcriptional activity of essential proteins involved in cell energy homeostasis. The integrative regulation of a number of proteins in the core of cell metabolism homeostasis affects the *BDNF* gene expression by various means, including its self-regulation via autocrine and/or paracrine TrkB signaling. BDNF/TrkB activation leads to the activation of several small G proteins in addition to the pathways regulated by mitogen-activated protein kinase (MAPK), PI 3-kinase (PI-3K), and phospholipase-Cγ (PLCγ) [218]. Meanwhile, as miR-101 suppresses MAPK phosphatases 1 (which dephosphorylates p38, JNK, and ERK), it has a positive effect on ERK phosphorylation and the downstream activation of *BDNF* expression in cortical neurons [98].

The activity of AMP-activated protein kinase (AMPK) and CREB represents the axis of cell energy metabolism. A compensatory increase in CREB activity following a decrease in the concentrations of BDNF and MeCP2 was evidenced in the brain of 132/212 KO mice [18,65]. While MeCP2 is a nuclear protein that may function as both a transcriptional activator or repressor, it works as a stabilizer of BDNF expression patterns and cell homeostasis [157,219,220]. Another compensatory effect is seen for BDNF in dendritogenesis when the inhibition of miR-15a, and the consequent relief of BDNF suppression, can rescue dendritic maturation deficits in MeCP2-deficient neurons [70]. Further, an upregulation in *BDNF* gene expression accompanies an increase in the expression of the miR-132/212 cluster, both of which target and suppress MeCP2 mRNA translation. The suppression of miR-132 and miR-212 on MeCP2 relieves its repression on *BDNF* expression. By this manner, the expression of BDNF and miR-132 and miR-212 represents a self-regulatory homeostatic mechanism that involves the nuclear protein MeCP2 at the core of cell metabolism [18,22,24,32,33,47,49,64,81,85,91].

The enzymatic activity of the histone deacetylase Sirtuin 1 (SIRT1) in the nicotinamide adenine dinucleotide (NAD)-dependent deacetylation of histones is crucial in protecting cells from oxidative stressors. SIRT1 activates the expression of mitochondrial DNA genes related to mitochondrial biogenesis, ATP generation, and cell proliferation. It was detected in experiments that SIRT1 is able to inhibit miR-134 expression by directly binding to its inhibitory elements, whereas SIRT1 deficiency and high levels of miR-134 result in a downregulation of CREB and *BDNF* expression, and a negative effect on neuronal survival/plasticity, another indirect mechanism by which miR-134 negatively affects BDNF function in the core of cell metabolism [37,71,72,149,160]. Finally, the study by Oikawa and colleagues [67] showed that the guanine nucleotide binding protein alpha inhibitor 1 (GNAI1), an adenylate cyclase inhibitor which regulates the ATP conversion to cAMP, is a target of miR-124. In physiological conditions, the suppression of GNAI1 by miR-124 increases in cAMP activity and leads to an upregulation of *BDNF* expression via the cAMP/PKA/CREB pathway. Indeed, alterations in cell metabolism and the microRNA environment reflect the regulation of BDNF.

MiR-124 suppression on BDNF activity negatively influences neuronal plasticity in various brain regions such as the hippocampus and striatum [21,35,46,77,100]. More recently, [148] identified that miR-124 targets CREB mRNA, consequently downregulating the *BDNF* expression, and alters BDNF function via targeting various gene transcripts’ downstream TrkB signaling, e.g., PI3K, Akt3, and Ras [181]. Likewise, the miR-124 negatively influences BDNF activity by targeting and degrading mRNA transcripts of glucocorticoid receptors [103,110]. Since signaling through glucocorticoid receptors potentiate BDNF actions via common intracellular pathways’ downstream TrkB receptors activation [221], such a suppression on glucocorticoid receptor synthesis might reflect a negative modulation of BDNF function. From another perspective, by testing different exercise intensities, Mojtahedi and colleagues [42] showed that miR-124 levels increase with the intensity, and this increase is amplified in strenuous intensity. BDNF and TrkB also increased with intensity, but not in the strenuous intensity exercise. The findings indicate a threshold beyond which the changes in metabolic demands evoke an acute rise in miR-124 levels and its suppressive effect overcoming that of BDNF.

Amongst the indirect effects seen for microRNA on BDNF [40], the study registered that miR-9 upregulates BDNF expression in retinal ganglion cells by suppressing the restrictive silencer factor/RE1-silencing transcription factor (REST), a transcription repressor whose suppression is required for neuronal cell differentiation. Similarly, miR-29c has a positive effect on BDNF expression levels by targeting DNA methyltransferase 3 [56]. The miR-705 was also found in a positive correlation with BDNF levels in an ischemic injured brain [106]. Further, BDNF administration increases miR-214 expression during embryonic stem cell differentiation into endothelial cells [119], and it is found to promote vascular endothelial growth factor-C-dependent lymph angiogenesis by suppressing miR-624-3p in human chondrosarcoma cells [97].

## 4. Conclusions

As a conservative protein with essential roles in the core of cell functioning, the secretion of BDNF, which modulates the expression and signal through TrkB receptors, is regulated by several systems, which, ultimately, protect cell growth and differentiation processes from oncogenicity. This means that precise control, whether endogenous or exogenous, in the availability of BDNF is necessary in order to achieve positive outcomes in neuroprotection and recovery. In this sense, our data collection indicates that multiple microRNAs co-operatively regulate BDNF and several core proteins responsible for cell growth and metabolism homeostasis, in neuronal survival and recovery. This provides a prospective insight for the development of vector-derived therapies that can potentially address and modulate BDNF locally and favor tissue damage recovery with a lower risk of oncogenesis.

## 5. Limitations

The regulation of BDNF by microRNAs involves a complex and dynamic regulation of basic proteins at the core of neuronal cell homeostasis. Further, the increasing evidence of the participation of different microRNAs in the regulation of BDNF leads us to assume that there shall be much to be uncovered about such integrated regulatory mechanisms involved in the direct and indirect regulation of BDNF by microRNAs. This complexity represents a relevant limitation to future research towards the development of therapeutic strategies.

## Figures and Tables

**Figure 1 cells-13-00880-f001:**
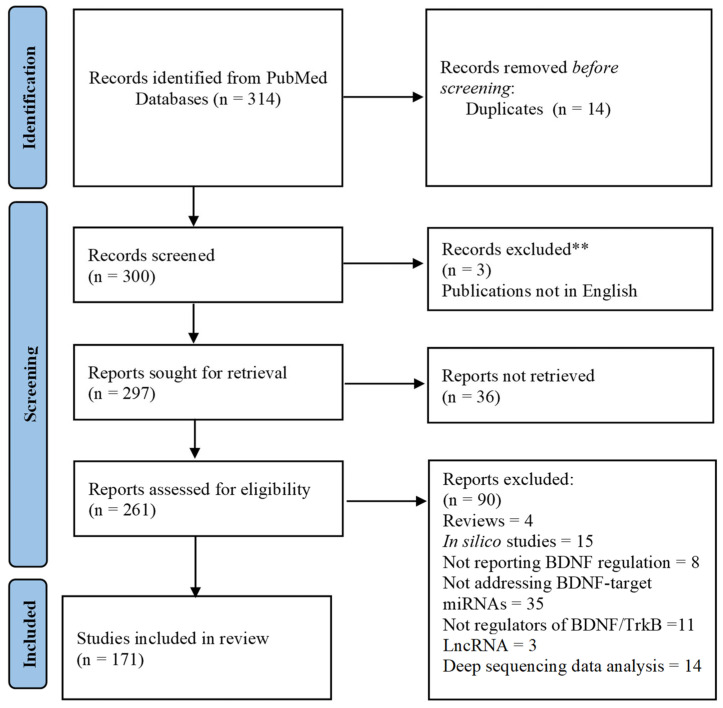
Search flowchart. ** Records excluded without assessment.

**Figure 2 cells-13-00880-f002:**
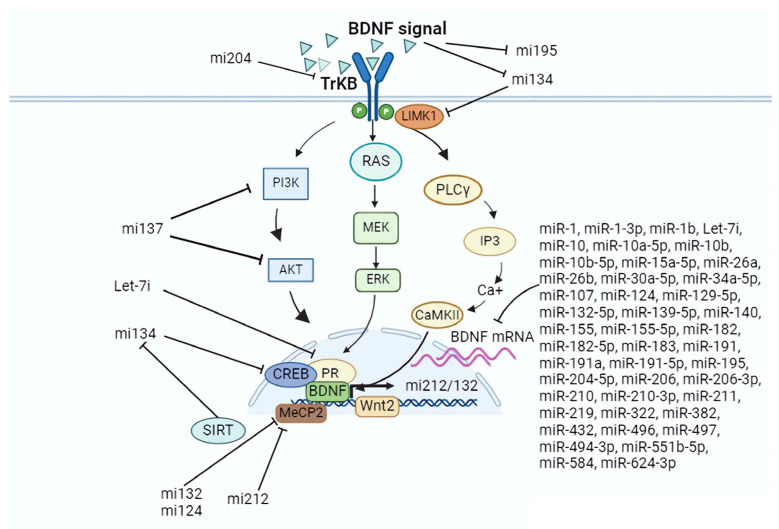
Experimental evidence of direct and indirect regulation of BDNF by microRNAs. Figure created by BioRender.com.

**Table 1 cells-13-00880-t001:** Main findings in BDNF regulation by microRNAs.

1	miR-134	BDNF inhibition of miR-134 favors synaptic plasticity	Schratt, G. M., et al. (2006) [17] https://doi.org/10.1038/nature04367
2	miR132	miR132 suppression on MeCP2 reduces BDNF levels in neurons	Klein, M. E., et al. (2007) [18] https://doi.org/10.1038/nn2010
3	miR-30a-5p miR-195	Inhibitors of BDNF in pre-frontal cortex	Mellios, N., et al. (2008) [19] https://doi.org/10.1093/hmg/ddn201
4			Mellios, N., et al. (2009) [20] https://doi.org/10.1016/j.biopsych.2008.11.019
5	miR-124	miR-124 suppresses BDNF levels in the brain upon cocaine administration	Chandrasekar, V., and Dreyer, J. L. (2009) [21] https://doi.org/10.1016/j.mcn.2009.08.009
6	MeCP2	miR-212 selectively suppresses the long MeCP2 mRNA form	Im, H.-I., et al. (2010) [22] https://doi.org/10.1038/nn.2615
7	miR-134	miR-134 suppresses CREB and BDNF impairing synaptic plasticity	Gao J., et al. (2010) [23] https://doi.org/10.1038/NATURE09271
8	miR-132	BDNF upregulates miR-132 expression in neurons	Kawashima, H., et al. (2010) [24] https://doi.org/10.1016/j.neuroscience.2009.11.057
9	miR-134	miR-134 suppression on LimK1 impaired BDNF-induced nerve growth	Han, L., et al. (2011) [25] https://doi.org/10.1186/1756-6606-4-40
10	miR-206	Shh signaling blocks miR-206 suppression on BDNF	Radzikinas, K., et al. (2011) [26] https://doi.org/10.1523/JNEUROSCI.2745-11.2011
11	miR-30a-5p	miR30a-5p suppresses BDNF translation in human glioblastoma-astrocytoma cell	Angelucci, F., et al. (2011) [27] https://doi.org/10.1159/000322528
12	miR-26	miR-26 suppresses BDNF translation in HeLa cells	Caputo, V., et al. (2011) [28] https://doi.org/10.1371/journal.pone.0028656
13	miR-16	miR-16 suppresses BDNF translation in the hippocampus	Bai, M., et al. (2012) [29] https://doi.org/10.1371/journal.pone.0046921
14	miR-206	miR-206 suppresses BDNF in the hippocampi of AD mice	Lee, S. T., et al. (2012) [30] https://doi.org/10.1002/ana.23588
15	miR-204	miR-204 suppression on BDNF control cancer cell migration and invasion	Imam, J. S., et al. (2012) [31] https://doi.org/10.1371/journal.pone.0052397
16	miR-132	BDNF upregulated miR-132/212 expression in neuronally differentiated SHSY5Y cells	Chen-Plotkin, A. S., et al. (2012) [32] https://doi.org/10.1523/JNEUROSCI.0521-12.2012
17	miR-132	BDNF upregulated miR-132 expression in hippocampal neurons	Wibrand, K., et al. (2012) [33] https://doi.org/10.1371/journal.pone.0041688
18	miR-206	miR-206 suppresses BDNF in skeletal muscle	Miura, P., et al. (2012) [34] https://doi.org/10.1111/j.1471-4159.2011.07583.x
19	miR-124a	miR-124a suppresses BDNF in rats’ brain	Bahi, A., & Dreyer, J.-L. (2013) [35] https://doi.org/10.1111/ejn.12228
20	miR-16	miR-16 suppression on BDNF regulates SHSY5Y cell growth	SUN, Y.-X., et al. (2013) [36] https://doi.org/10.3892/or.2013.2731
21	miR-132 miR-182	Serum miR-132 and miR-182 levels negatively correlate with BDNF’s in patients with depression	Li, Y.-J., et al. (2013) [37] https://doi.org/10.1371/journal.pone.0063648
22	miR-124a	The ‘pain-relevant’ miRNA-124a targets MeCP2 in spinal cord	Kynast, K. L., et al. (2013) [38] https://doi.org/10.1016/j.pain.2012.11.010
23	miR-30a-5p	miR-30a-5p suppresses BDNF in rat cortical neurons	Croce, N., et al. (2013) [39] https://doi.org/10.1007/s11010-013-1567-0
24	miR-212	Electroconvulsive therapy increases BDNF and miR-212 in rats’ brain	Ryan, K. M., et al. (2013) [40] https://doi.org/10.1016/j.neulet.2013.05.035
25	miR-191	miR-191 suppresses BDNF in human breast cancer cells	Nagpal, N., et al. (2013) [41] https://doi.org/10.1093/carcin/bgt107
26	miR-124	miR-124 increases with exercise intensity	Mojtahedi, S., et al. (2013) [42] https://doi.org/10.1002/cbin.10022
27	miR-206	miR-206 suppresses BDNF in rat medial prefrontal cortex	Tapocik, J. D., et al. (2014) [43] https://doi.org/10.1523/JNEUROSCI.0445-14.2014
28	miR-183	miR-183 negatively correlates with BDNF in the dorsal root ganglion	Lin, C. R., et al. (2014) [44] https://doi.org/10.1111/ejn.12522
29	miR-206	miR-206 suppresses BDNF in hippocampal tissue	Tian, N., et al. (2014) [45] https://doi.org/10.1007/s12264-013-1419-7
30	miR-124a	miR-124a suppressed BDNF in the hippocampus of rodent exposed to social defeat stress	Bahi, A., et al. (2014) [46] https://doi.org/10.1016/j.psyneuen.2014.04.009
31	miR-132	BDNF promotes axon branching of retinal ganglion cells via upregulation of miR-132 targeting p250GAP	Marler, K. J., et al. (2014) [47] https://doi.org/10.1523/JNEUROSCI.1910-13.2014
32	miR-206	miR-206 targeting of BDNF in hippocampal cells was attenuated by ketamine	Yang, X., et al. (2014) [48] https://doi.org/10.1007/s12017-014-8312-z
33	miR-132	Activation of ERK/CREB is associated with miR-132 expression and hippocampal neuronal proliferation	Yi, L. T., et al. (2014) [49] https://doi.org/10.1503/jpn.130169
34	miR-124 miR-132 Let7d	let7d, miR-124, and miR-132 were negatively associated with BDNF in the brain of rats exposed to cocaine	Giannotti, G., et al. (2014) [50] https://doi.org/10.1017/S1461145713001454
35	miR-1 miR-10b miR-155 miR-191	miR-1, -10b, -155, and miR-191 directly target BDNF in various human cell cultures	Varendi, K., et al. (2014) [51] https://doi.org/10.1007/s00018-014-1628-x
36	miR-34a	Hypoxia caused a decrease in serum BDNF and miR-34a expression in the lower brainstem	Zhang, J., et al. (2014) [52] https://doi.org/10.3760/cma.j.issn.0366-6999.20131683
37	miR-let-7a	miR-let-7a suppresses the expression of inducible nitric oxide synthase (iNOS), IL-6, favoring BDNF expression	Cho, K. J., et al. (2015) [53] https://doi.org/10.1016/j.mcn.2015.07.004
38	miR-183/96/182	miR 183/96/182 cluster targets BDNF transcripts	Li, H., et al. (2015) [54] https://doi.org/10.3892/mmr.2015.3736
39	miR-1	miR-1 suppresses BDNF in heart and hippocampal tissues	Ma, J. C., et al. (2015) [55] https://doi.org/10.1016/j.neuroscience.2015.04.061
40	miR-29c	miR-29c is found to be positively correlated with BDNF in the cerebral fluid of AD patients	Yang, G., et al. (2015) [56] https://doi.org/10.3892/mmr.2015.3531
41	miR-1	Chronic constriction injury leads to a decrease in miR-1 with a consequent increase in BDNF	Neumann, E., et al. (2015) [57] https://doi.org/10.1186/s12990-015-0045-y
42	miR-22	miR-22 negatively correlates with BDNF in human neuroblastoma cells treated with Perfluorooctane sulfonate	Li, W., et al. (2015) [58] https://doi.org/10.1155/2015/302653
43	miR-132	miR-132 aggravates epileptiform discharges in cultured hippocampal neurons via BDNF suppression	Xiang, L., et al. (2015) [59] https://doi.org/10.1016/j.brainres.2015.06.046
44	miR-204	miR-204 decreases BDNF expression and invasive and metastatic behavior in epithelial ovarian cancer cells	Yan, H., et al. (2015) [60] https://doi.org/10.1097/IGC.0000000000000456
45	miR-206-3p	miR-206-3p suppresses BDNF in mouse skin development	Mu, Y., et al. (2015) [61] https://doi.org/10.3892/mmr.2015.4456
46	miR-134 miR-132	BDNF upregulates Limk1 translation and phosphorylation via modulation of miR-134 and miR-132	Li, M., et al. (2015) [62] https://doi.org/10.1002/path.4484
47	miR-937	Transplantation of antisense-miR-937-expressing mesenchymal cells increased BDNF levels in AD mice	Liu, Z., et al. (2015) [63] https://doi.org/10.1159/000430356
48	miR-132	The levels of MeCP2 and BDNF negatively correlate with those of miR-132 in patients with major depressive disorder	Su, M., et al. (2015) [64] https://doi.org/10.3892/mmr.2015.4104
49	miR-132/212	miR-132/212 knockout mice present a marked decrease of MeCP2 and BDNF levels in the hippocampus, and an increase in phosphorylated CREB	Hernandez-Rapp, J., et al. (2015). [65] https://doi.org/10.1016/j.bbr.2015.03.032
50	miR-134	AMPK has a negative effect on total CREB expression by elevating SIRT1/miR-134	Huang, W., et al. (2015) [66] https://doi.org/10.1007/s12031-015-0500-2
51	miR-124	miR-124 suppression on guanine nucleotide binding protein alpha inhibitor 1 (GNAI1) increases	Oikawa, H., et al. (2015) [67] http://doi.org/10.1016/j.neuint.2015.10.010
52	miR-10B	BDNF was identified as a direct target gene of miR-10B in rats	Jiang, Y., and Zhu, J. (2015) [68] http://www.ncbi.nlm.nih.gov/pubmed/25755749
53	miR-134	miR-134 inhibition elevated the expression of CREB and BDNF in retinal ganglion cell	Shao, Y., et al. (2015) [69] https://doi.org/10.1007/s12031-015-0522-9
54	miR-15a	miR-15a suppresses BDNF and neuronal maturation	Gao, Y., et al. (2015) [70] https://doi.org/10.1002/stem.1950
55	miR-30a-5p	miR-30a-5p suppresses BDNF expression in the medial prefrontal cortex	Darcq, E., et al. (2015) [71] https://doi.org/10.1038/mp.2014.120
56	miR-15a-5p	miR-15a-5p suppresses BDNF expression in human hepatocellular carcinoma	Long, J., et al. (2016) [72] https://doi.org/10.1007/s13277-015-4427-6
57	miR-219	miR-219 suppresses CaMKIIγ and, consequently, enhances BDNF production in mouse dorsal root ganglia	Hu, X. M., et al. (2016) [73] https://doi.org/10.1177/1744806916666283
58	miR-182	miR-182 upregulation correlated with a decrease in BDNF expression in the hippocampus of rats with chronic unpredictable mild stress	Li, Y., et al. (2016) [74] https://doi.org/10.1016/j.pnpbp.2015.09.004
59	miR-10b	miR-10b suppresses goat granulosa cell proliferation by targeting BDNF	Peng, J. Y., et al. (2016) [75] https://doi.org/10.1016/j.domaniend.2015.09.005
60	miR-1	miR-1 targeting BDNF regulates Schwann cell proliferation and migration after peripheral nerve injury	Yi, S., et al. (2016) [76] https://doi.org/10.1038/srep29121
61	miR-124a	Neonatal isolation-inducible cognitive impairments lead to induction of miR124a and suppression on BDNF in rat	Bahi, A. (2016) [77] https://doi.org/10.1016/j.bbr.2016.05.033
62	miR-107	BDNF is a direct target of miR-107 in non-small-cell lung cancer cells	Xia, H., Li, Y., and Lv, X. (2016) [78] https://doi.org/10.3892/ijo.2016.3628
63	miR-1	Deletion of *Bdnf* in dorsal root ganglion neurons leads to a temporary dysregulation of miR-1	Neumann, E., et al. (2016) [79] https://doi.org/10.1016/j.mcn.2016.06.003
64	miR-132 miR-134	BDNF acts in concert with Limk-1, miR-132, and miR-134 for the regulation of structural and morphological plasticity	Kumari, A., et al. (2016) [80] https://doi.org/10.1016/j.physbeh.2016.02.032
65	miR-212/132	Intrathecal Ad-CRTC1 downregulated the expression of miRNA-212/132, p-CREB, and BDNF in spinal cord in tumor-bearing mice	Liang, Y., et al. (2016) [81] https://doi.org/10.1177/1744806916641679
66	miR-9	miR-9 suppression on the transcriptional repressor RE1-silencing transcription factor favors BDNF expression in mouse retinal ganglion cells.	Jiang, B., et al. (2016) [82] https://doi.org/10.3892/mmr.2016.5810
67	miR-195	BDNF-mediated downregulation of miR-195 inhibits ischemic cardiac apoptosis in rats	Hang, P., et al. (2016) [83] https://doi.org/10.7150/ijbs.15071
68	miR-613	miR-613 is found to be negatively correlated with BDNF in serum, cerebrospinal fluid, and hippocampus of patients with AD.	Li, W., et al. (2016) [84] https://doi.org/10.5582/bst.2016.01127
69	miR-212 miR-132	miR- 212/132 regulates pattern changes and *Bdnf* through inhibition of MeCP2	Jimenez-Gonzalez, A., et al. (2016) [85] https://doi.org/10.1016/j.bbagen.2016.03.001
70	miR-10b	miR-10b suppresses the migration and invasion of chondrosarcoma cells by targeting BDNF	Aili, A., Chen, Y., and Zhang, H. (2016) [86] https://doi.org/10.3892/mmr.2015.4506
71	miR-210	miR-210 upregulation increased mBDNF/proBDNF ratio in normal and ischemic mouse brain	Zeng, L. L., et al. (2016) [87] https://doi.org/10.1111/cns.12589
72	miR-204	miR-204 suppresses TrkB in cultured hippocampal neurons	Xiang, L., et al. (2016) [88] https://doi.org/10.1016/j.brainres.2016.02.045
73	miR-34a-5p	Total abdominal irradiation elevates miR-34a-5p in the intestine, resulting in reduction of hippocampal BDNF	Cui, M., et al. (2017) [89] https://doi.org/10.1016/j.bbadis.2017.06.021
74	miR-137	miR-137 targets proteins in the PI3K-Akt-mTOR pathway	Thomas, K. T., et al. (2017) [90] https://doi.org/10.1016/j.celrep.2017.06.038
75	miR-132/212	Suprachiasmatic nucleus neurons from miR-132/212-deficient mice have reduced dendritic spine density, along with altered MeCP2 and BDNF	Mendoza-Viveros, L., et al. (2017) [91] https://doi.org/10.1016/j.celrep.2017.03.057
76	miR-206	Serum miR-206 is a biomarker of Alzheimer’s disease	Xie, B., et al. (2017) [92] https://doi.org/10.3233/JAD-160468
77	miR-497	miR-497 inhibits thyroid cancer tumor growth and invasion by suppressing BDNF	Wang, P., et al. (2017) [93] https://doi.org/10.18632/oncotarget.13747
78	miR-107	miR-107 has a suppressive effect in breast cancer by negatively regulating BDNF	Gao, B., et al. (2017) [94] https://doi.org/10.1002/jgm.2932
79	miR-140	miR-140 suppresses BDNF expression in astrocytes	Tu, Z., et al. (2017) [95] https://doi.org/10.1016/j.biopha.2017.05.016
80	miR-382	miR-382 inhibits cell proliferation and invasion of retinoblastoma by targeting BDNF	Song, D., et al. (2017) [96] https://doi.org/10.3892/mmr.2017.7396
81	miR-624-3p	miR-624-3p expression was negatively regulated by BDNF via the MEK/ERK/mTOR cascade	Lin, C. Y., et al. (2017) [97] https://doi.org/10.1038/cddis.2017.354
82	miR-101	miR-101 suppresses dual specific phosphatase 1 expression and inhibited the downstream BDNF expression	Zhao, Y., et al. (2017) [98] https://doi.org/10.1016/j.brainres.2017.05.020
83	miR-211	miR-211 suppresses BDNF expression in human astrocytes	Zhang, K., et al. (2017) [99] https://doi.org/10.1042/BSR20170755
84	miR124a	Hippocampal miR-124a silencing or BDNF overexpression attenuated anxiety- and autism-like behaviors in rats	Bahi, A. (2017) [100] https://doi.org/10.1016/j.bbr.2017.03.010
85	miR-744	miR-744 inhibits tumor cell proliferation and invasion of gastric cancer via suppression of BDNF	Xu, A. J., et al. (2017) [101] https://doi.org/10.3892/mmr.2017.7167
86	miR-206	miR-206 ameliorates chronic constriction injury-induced neuropathic pain in rats via suppression on BDNF	Sun, W., et al. (2017) [102] https://doi.org/10.1016/j.neulet.2016.12.047
87	miR-124	miR-124 suppression on GR has a negative effect on BDNF-TrkB signaling pathway in the hippocampus	Wang, S. S., et al. (2017) [103] https://doi.org/10.1016/j.pnpbp.2017.07.024
88	miR-103	miR-103 inhibits glioma cell proliferation and invasion by suppressing BDNF	Wang et al., 2017 [104] https://doi.org/10.3892/mmr.2017.8282
89	MiR-134	BDNF inhibits MiR-134 expression by activating the TrkB pathway	Huang, W., et al. (2017) [105] https://doi.org/10.1007/s12031-017-0907-z
90	miR 705	miR-705 overexpression mitigates neurological deficits in ischemic brain damage	Ji, M., et al. (2017) [106] https://doi.org/10.3892/mmr.2017.7626
91	miR-125b-5p	BDNF expression is negatively regulated by miR-125b-5p in rod bipolar cells under degeneration	Fu et al., (2017) [107] http://dx.doi.org/10.1038/s41598-017-01261-x
92	miR-221	miR-221 suppresses Wnt2 and, consequently, p-CREB and BDNF expression in hippocampal neurons	Lian, N., et al. (2018) [108] https://doi.org/10.1080/15384101.2018.1556060
93	miR-155	Overexpression of miRNA-155 resulted in decreased BDNF and TrkB protein expression in epilepsy cells	Duan, W., et al. (2018) [109] https://doi.org/10.3892/ijmm.2018.3711
94	miR-124	Reduction in miR-124 suppression on GR and BDNF was required for the antidepressant-like effects of gypenosides induced by chronic corticosterone injection in mice	Yi, L. T., et al. (2018) [110] https://doi.org/10.1177/0269881118758304
95	let-7i	Inhibition of let-7i suppression on progesterone receptor membrane component 1 and BDNF enhances progesterone’s protective effects against stroke	Nguyen, T., et al. (2018) [111] https://doi.org/10.1073/pnas.1803384115
96	miR-107	miR-107 acts as tumor inhibitor for gastric cancer through targeting BDNF expression in gastric cancer cells	Cheng, F., et al. (2018) [112] https://doi.org/10.1016/j.micpath.2018.04.060
97	miR-132	Plasma BDNF levels are increased in patients with major depressive disorder, and miR-132 correlates with anxiety and depression symptoms	Fang, Y., et al. (2018) [113] https://doi.org/10.1016/j.jad.2017.11.090
98	miR-210-3p	Inhibition of BDNF production upregulation of miR-210-3p contributes to dopaminergic neuron damage in MPTP model	Zhang, S., et al. (2018) [114] https://doi.org/10.1016/j.neulet.2017.10.014
99	miR-206	miR-206 is a post-transcriptional inhibitor of BDNF in pregnant hypothyroid rats	Xing, Q., et al. (2018) [115] https://doi.org/10.1055/a-0658-2095
100	miR-497	BDNF was found to be negatively regulated by miR-497 and associated with the apoptosis of Müller cells under high glucose	Li, X. J. (2018) [116] https://doi.org/10.1177/1479164117749382
101	miR-155	Minocycline is neuroprotective against ischemic brain injury through their modulation of miR-155-mediated BDNF repression	Lu, Y., et al. (2018) [117] https://doi.org/10.1007/s10571-018-0599-0
102	miR-206-3p	Stress-induced mood alterations in pregnant mice correlate with changes in miR-206-3p and BDNF expression in the hippocampus and amygdala	Miao, Z., et al. (2018) [118] https://doi.org/10.1007/s12035-016-0378-1
103	miR-214	miR-214 mediated the BDNF-induced expressional changes in embryonic stem cells, contributing to BDNF-driven endothelial differentiation	Descamps, B., et al. (2018) [119] https://doi.org/10.1161/ATVBAHA.118.311400
104	miR-26a miR-125b	Upregulation of BDNF is associated with reduced miR-26a and miR-125b in APP/PS1 mice under vitamin D treatment	Lv, M., et al. (2018) [120] https://doi.org/10.1002/mnfr.201800621
105	miR-132 miR-204	Increases in miR-132 and miR-204 and decrease in BDNF expression are found in the hippocampus of rats exposed to fluorine/aluminium.	Ge, Q. Di, et al. (2018) [121] https://doi.org/10.1016/j.etap.2018.08.011
106	miR-134	Resveratrol treatment increases Sirt1, p-CREB, CREB, and BDNF expression and decreases miR134 levels in hippocampus	Shen, J., et al. (2018) [122] https://doi.org/10.1016/j.bbr.2018.04.050
107	miR-10a	miR-10a suppresses BDNF expression in rats with AD	Wu, B. W., et al. (2018) [123] https://doi.org/10.1002/jcp.26328
108	miR-1	Inhibition of miR-1 suppression on BDNF in the hippocampus ameliorates myocardial infarction induced impairment of long-term potentiation	Ma, J. C., et al. (2018) [124] https://doi.org/10.1159/000494657
109	miR-10a	miR-10a suppression on BDNF controls airway smooth muscle cell proliferation in asthma	Zhang, X. Yu, et al. (2018) [125] https://doi.org/10.1016/j.lfs.2018.09.002
110	MiR-1-3p	miR-1-3p suppression on BDNF regulates viability, proliferation, invasion, and apoptosis of bladder cancer cells	Gao, L., et al. (2018) [126] https://doi.org/10.4149/neo_2018_161128N594
111	miR-322	miR-322 suppression on BDNF promotes Tau phosphorylation in AD mouse brain	Zhang, J., et al. (2018) [127] https://doi.org/10.1007/s11064-018-2475-1
112	miR-107	Ketamine induces neural injury via miR-107 suppression on BDNF in embryonic-stem-cell-derived neurons	Jiang, J. D., et al. (2019) [128] https://doi.org/10.1002/iub.1911
113	miR-124	miR-124 improved rats’ spatial learning and memory ability and hippocampal neuron viability and resistance to apoptosis, corresponding to an increased BDNF expression	Yang, W., et al. (2019) [129] https://doi.org/10.1002/jcp.28862
114	miR-206	miR-206 has the potential to specifically regulate BDNF with a long 3′ UTR without affecting its short 3′ UTR counterpart	Shrestha, S., et al. (2019) [130] https://doi.org/10.1002/2211-5463.12581
115	miR-30a	Presence of the pregnant partner regulates miR-30a suppression on BDNF and protects male mice from social-defeat-induced abnormal behaviors	Miao, Z., et al. (2019) [131] https://doi.org/10.1016/j.neuropharm.2019.03.032
116	miR-584	miR-584 suppression on BDNF inhibits hepatocellular carcinoma cell proliferation and invasion	Song, Y., et al. (2019) [132] https://doi.org/10.3892/mmr.2019.10424
117	miR-497	miR-497 targets BDNF in papillary thyroid carcinoma	Sun, Z., et al. (2019) [133] https://doi.org/10.1002/jcp.26928
118	miR-191a	miR-191a showed negative correlation with BDNF in ovariectomized rats in sleep deprivation	Mohammadipoor-Ghasemabad, L., et al. (2019) [134] https://doi.org/10.1016/j.neuroscience.2019.06.037
119	miR-204-5p	miR-204-5p suppression on BDNF expression influence on the depressive-like behaviors in mice under the chronic mild stress	Hung, Y. Y., et al. (2019) [135] https://doi.org/10.3390/cells8091021
120	miR-7	miR-7 suppresses BDNF and α-synuclein axis in Parkinson’s disease	Li, B. B., et al. (2019) [136] https://doi.org/10.1016/j.chemosphere.2019.05.064
121	miR-496	miR-496 suppression on BDNF controls non-small-cell lung cancer growth	Ma, R., et al. (2019) [137] https://doi.org/10.1016/j.bbrc.2019.08.046
122	miR-134	SIRT1/miR-134 signaling pathway regulates BDNF expression in primary cultured hippocampal neurons	Shen, J., et al. (2019) [138] https://doi.org/10.1016/j.jad.2019.01.031
123	miR-206	Chronic ethanol, stress, and their combination alter miR-206 suppression on BDNF in brain	Solomon, M. G., et al. (2019) [139] https://doi.org/10.1016/j.neuroscience.2019.02.012
124	miR-375	Inhibition of miR-375 ameliorates ketamine-induced neurotoxicity and BDNF expression in neurons	Zhao, X., et al. (2019) [140] https://doi.org/10.1016/j.ejphar.2018.11.035
125	miR-363-3p	miR-363-3p attenuates depressive-like behaviors and elevates BDNF levels	Panta, A., et al. (2019) [141] https://doi.org/10.1016/j.bbi.2019.01.003
126	miR-206	Inhibition of miR-206 suppression on BDNF improves neurological deficit and brain edema and suppresses neuronal apoptosis in subarachnoid hemorrhage	Zhao, H., et al. (2019) [142] https://doi.org/10.1016/j.neuroscience.2019.07.051
127	miR-185	miR-185 suppression on truncated TrkB receptors activates full-length TrkB signaling and reduces epileptiform discharges in cultured hippocampal neurons	Xie, W., et al. (2020) [143] https://doi.org/10.1007/s11064-020-03013-2
128	miR-494-3p	Pramipexole inhibits MPP+-induced neurotoxicity by miR-494-3p suppression on BDNF	Deng, C., et al. (2020) [144] https://doi.org/10.1007/s11064-019-02910-5
129	miR-15a	miR-15a suppression on BDNF exerts a negative regulatory effect on the oxygen-glucose deprivation/reoxygenation injury.	Hu, J. J., et al. (2020) [145] https://doi.org/10.1002/kjm2.12136
130	miR-192-5p	MiR-192-5p inhibition inhibited neuronal apoptosis by affecting the expression of BDNF	Liu, X., et al. (2020) [146] https://doi.org/10.1080/15384101.2019.1710916
131	miR-10a-5p	Inhibition of miR-10a-5p suppression on BDNF enhances the therapeutic effect on spinal cord in injury bone marrow mesenchymal stem cells	Zhang, T., et al. (2020) [147] https://doi.org/10.1016/j.neulet.2019.134562
132	miR-124	Knockdown of miR-124 reduces depression-like behavior by suppression on CREB and BDNF	Yang, W., et al. (2020) [148] https://doi.org/10.2174/1567202617666200319141755
133	miR-129-5p	Metastasis-associated lung adenocarcinoma transcript 1 promotes Schwann cell proliferation and migration by reducing miR-129-5p suppression on BDNF	Wu, G., et al. (2020) [149] https://doi.org/10.1016/j.yexcr.2020.111937
134	miR-204-5p	miR-204-5p mediates sevoflurane-induced cytotoxicity in hippocampal cells by targeting BDNF	Liu, H., et al. (2020) [150] https://doi.org/10.14670/HH-18-266
135	miR-10a-5p	miR-10a-5p suppresses BDNF and neuronal growth in Friedreich’s ataxia	Misiorek, J. O., et al. (2020) [151] https://doi.org/10.1007/s12035-020-01899-1
136	mir-210	miR-210 suppression on BDNF participates in mesenchymal-stem-cell-modulated neural precursor cell migration	Wang, F., et al. (2020) [152]. https://doi.org/10.3892/mmr.2020.11065
137	miR-10b-5p	Dexmedetomidine has neuroprotective effects on hippocampal neuronal cells via regulation of miR-10b-5p suppression on BDNF	Wang, L., et al. (2020) [153] https://doi.org/10.1007/s11010-020-03726-6
138	miR-103-3p	LncRNA BC083743 promotes Schwann cell proliferation and axon regeneration via miR-103-3p suppression on BDNF after sciatic nerve crush	Gao, L., et al. (2020) [154] https://doi.org/10.1093/JNEN/NLAA069
139	miR-195-5p	Serum miR-195-5p and miR-451a levels inversely correlate with those of BDNF in stroke patients	Giordano, M., et al. (2020) [155] https://doi.org/10.3390/ijms21207615
140	miR-216a-5p	BDNF addition to exosome-derived therapy improves recovery after traumatic brain injury via increasing miR-216a-5p expression	Xu, H., et al. (2020) [156] https://doi.org/10.12659/MSM.920855
141	miR-132	miR-132 (both miR132-3p and miR132-5p) and BDNF transcripts are significantly lower in Rett syndrome patients	Pejhan et al., (2020) [157] https://doi.org/10.3389/fcell.2020.00763
142	miR-155	Inhibition of miR-155 suppression on BDNF reduces cardiomyocyte apoptosis	Lin, B., et al. (2021) [158] https://doi.org/10.18632/aging.103640
143	miR-186	Aerobic exercise reduces miR-186 suppression on BDNF and neuronal apoptosis in vascular cognitive impairment	Niu, Y., et al. (2021) [159] https://doi.org/10.1186/s10020-020-00258-z
144	miR-432	Adenosine deaminase acting on RNA1 alleviates the depressive-like behavior via regulation of miR-432 suppression on BDNF	Zhang, X., et al. (2021) [160] https://doi.org/10.1016/j.bbr.2020.113087
145	miR-155	lncRNA MIR155HG alleviates depression-like behaviors in mice by regulating miR-155 suppression on BDNF	Huan, Z., et al. (2021) [161] https://doi.org/10.1007/s11064-021-03234-z
146	miR 365	miR-365 suppresses BDNF in streptozotocin-induced diabetic nephropathy fibrosis and renal function	Zhao, P., et al. (2021) [162] https://doi.org/10.1007/s11255-021-02853-3
147	miR-10b-5p	Changes in miR-10b promoter may contribute to upregulation of miR-10b-5p suppression on BDNF and hippocampal neurogenesis and cognition in mice	Ke, X., et al. (2021) [163] https://doi.org/10.1159/000515750
148	miR-191-5p	Long non-coding RNA XIST promotes retinoblastoma cell proliferation, migration, and invasion by modulating miR-191-5p suppression on BDNF	Xu, Y., et al. (2021) [164] https://doi.org/10.1080/21655979.2021.1918991
149	miR-195	Higher miR-195 expression was significantly correlated to lower BDNF in levels and poorer overall cognitive performance in schizophrenia patients	Pan, S., et al. (2021) [165] https://doi.org/10.1038/s41398-021-01240-x
150	miR-191	Inhibition of miR-191 suppression on BDNF protects against isoflurane-induced neurotoxicity	Li, H., et al. (2021) [166] https://doi.org/10.1080/15376516.2021.1886211
151	miR-155	Chronic colitis impairs heart function through miR-155 suppression on BDNF	Tang, Y., et al. (2021) [167] https://doi.org/10.1371/journal.pone.0257280
152	miR-1b	Upregulation of miR-1b suppression on BDNF reduces neuron viability and regenerative ability	Li, X., et al. (2021) [168] https://doi.org/10.5114/fn.2021.105132
153	miR-206-3p	BDNF was negatively regulated by miR-206-3p in AD mice	Peng, D., et al. (2022) [169] https://doi.org/10.7150/THNO.70951
154	miR-191-5p	miR-191-5p disturbed angiogenesis in a mice model of cerebral infarction by suppressing BDNF	Wu, Y., et al. (2021) [170] https://doi.org/10.4103/0028-3886.333459
155	miR-206-3p	miR-206-3p suppression on hippocampal BDNF participates in the pathogenesis of depression	Guan, W., et al. (2021) [171] https://doi.org/10.1016/j.phrs.2021.105932
156	miR-103a-3p miR-10a-5p	miR-103a-3p or miR-10a-5p negatively affects the maturation of oocytes by suppressing BDNF in follicular fluid	Zhang, Q., et al. (2021) [172] https://doi.org/10.3389/fendo.2021.637384
157	miR30a-5p miR-195-5p miR191-5p miR206-3p	Increased expression of miR30a-5p, miR-195-5p, miR191-5p, and miR206-3p was detected in the rapid drinking onset rats	Ehinger, Y., et al. (2021) [173] https://doi.org/10.1111/adb.12890
158	miR-182-5p	Serum miR-182-5p was elevated and BDNF expression was lowered in chronic heart failure patients	Fang, F., et al. (2022) [174] https://doi.org/10.1186/s13019-022-01802-0
159	miR-139-5p	miR-139-5p inhibition plays an antidepressant-like role via suppression on BDNF	Su, B., et al. (2022) [175] https://doi.org/10.1080/21655979.2022.2059937
160	miR-497	miR-497 suppression on BDNF impairs the proliferation, migration, and oxidative stress response of Schwann cells	Yongguang, L., et al. (2022) [176] https://doi.org/10.1186/s12906-021-03483-z
161	miR-206-3p	Electroacupuncture alleviates neuropathic pain after chronic constriction injury via miR-206-3p suppression on BDNF	Tu, W., et al. (2022) [177] https://doi.org/10.1155/2022/1489841
162	miR-155-5p	Triptolide inhibits miR-155-5p suppression on BDNF and reduces podocyte injury in mice with diabetic nephropathy	Gao, J., et al. (2022) [178] https://doi.org/10.1080/21655979.2022.2067293
163	miR-210	Inhibition of miR-210 resulted in increased viability and reduced apoptosis, along with increased BDNF levels after hypoxia/reoxygenation	Zhai, Y., et al. (2022) [179] https://doi.org/10.1002/kjm2.12486
164	miR-3168	BDNF upregulated the expression of miR-3168 in macrophages	Yu, H. C., et al. (2022) [180] https://doi.org/10.3390/ijms23010570
165	miR-124-3p	miR-124-3p inhibition promotes subventricular zone neural stem cell activation by enhancing BDNF function after traumatic brain injury in adult rats	Kang, E. M., et al. (2022) [181] https://doi.org/10.1111/cns.13845
166	miR-182-5p	miR-182-5p suppression on BDNF and angiogenin affects retinal neovascularization	Li, C., et al. (2022) [182] https://doi.org/10.3892/mmr.2021.12577
167	miR-210-3p	miR-210-3p suppresses osteogenic differentiation of osteoblast precursor cell by targeting BDNF	Deng, L., et al. (2022) [183] https://doi.org/10.1186/s13018-022-03315-x
168	miR-1-3p	High miR-1-3p expression and low serum BDNF levels were found in patients with primary hypertension complicated with depression	Ding, J., Jiang, C., Yang, L., and Wang, X. (2022) [184] https://doi.org/10.14715/CMB/2022.68.1.10
169	miR-132-5p	miR-132-5p suppression on BDNF in the prefrontal cortex resulted in depression-like behaviors	Ma, L., et al. (2022) [185] https://doi.org/10.1038/s41398-022-02192-6
170	miR-551b-5p	miR-551b-5p suppression on BDNF participates in early convalescence by intermittent theta burst stimulation	Wang, L., et al. (2022) [186] https://doi.org/10.1016/j.brainresbull.2022.03.002
171	miR-382 miR-182	miR-382/miR-182 suppression on BDNF have a positive effect in the management of post-stroke depression	Zhang, Z., et al. (2022) [187] https://doi.org/10.1016/j.bbrc.2022.05.038

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
