# Peer review of "BDNF Modulation by microRNAs: An Update on the Experimental Evidence"

_cells, 2024, doi:10.3390/cells13100880_

Round 1

Reviewer 1 Report

Comments and Suggestions for Authors

In this manuscript, Assis and Murawska-Ciałowicz provide an overview of how microRNAs regulate BDNF, involving dynamic control of key proteins that are central to cellular homeostasis. They emphasize that local BDNF regulation by microRNAs entails a complex interplay between multiple microRNAs and target proteins that can either suppress or enhance BDNF expression, influencing core cellular metabolic processes. This review offers a comprehensive examination of how microRNAs modulate BDNF and suggests that utilizing multiple microRNAs to cooperatively influence BDNF function could be a promising strategy for future vector-delivery-based treatments.

However, I have three minor concerns that, if addressed, would enhance the manuscript:

The authors conducted a review of 173 papers on 'BDNF and MicroRNA,' but the abstract states that 171 papers were included for review.

Certain abbreviations, such as 'methyl CpG binding protein 2,' should be spelled out upon their initial mention in the main text.

The underlined words should be removed, and consistency should be maintained in terms of formatting for words such as 'histones' in line 264 and 'mitochondrial biogenesis' in line 266.

Author Response

Please, find the referred revision attached, and changes in the manuscript marked in blue.

Reviewer 2 Report

Comments and Suggestions for Authors

This review is a literature search of small interfering RNA modulating BNDF activation or expression.  The study examines 172 publications and organizes the discussion into 3 categories, BDNF by microRNA, neuroplasticity and BDNF regulation by microRNAs and Cell metabolism and BDNF regulation.  However, many of the microRNAs have an indirect on BDNF function by effecting general growth pathways and factors regulated by multiple neurotrophins.  It is unclear how specific many of these microRNAs are for BDNF pathways or growth pathways in general.  It would be best for the authors to focus on those more specifically regulating BDNF function.  Additionally figure 2 is very confusing with multiple lines everywhere, it is often unclear how these pathways are connected to effect BDNF modulation.   Finally, there are numerous grammatic errors and confusingly written sentences. 

Comments:

1)      Redo figure 2 to better illustrate those miRs directly effecting BDNF regulation and those effecting indirect and to reduce and clarify the lines within the figure.  It would be best if the figure was rendered down into several panels describing each of the processes the authors are attempting to illustrate.  It would also be helpful for the authors to separately show direct and indirect effect of miRs on BDNF regulation and better illustrate the biochemical pathways leading to BDNF modulation by these indirect pathways.

2)      It would also be informative for the authors to go into further details about which miRs directly and indirectly effect BDNF, in both the list of authors and discussion.

3)      Organization of table 1 is also confusing and hard to follow when reading the discussion.  The authors should first list the miR, then whether the paper describes BDNF role in cancer, development, learning, pain, plasticity, etc., flowed by the citation last.   In present form table 1 adds nothing to the overall understanding of the manuscript except for the overall page count.

Comments on the Quality of English Language

Overall the majority of sentence are written well, however, there are many that are not.  mostly small items, e.g,

line 285: Likewise, miR- 284 124 negatively influence on BDNF signal...

line 290: but not in strenuous the intensity exercise.

line 108: microRNAs can be are exported....

lines 264 and 266: the words histones and mitochondrial biogenesis are different font, font size (much bigger than the rest of the text) and underlined, why? 

and many other 

Author Response

Please, find the referred revision attached, and changes in the manuscript marked in green.

Reviewer 3 Report

Comments and Suggestions for Authors

I have following comments:

Table one is unnecessary and recommend removing it.

"Neuroplasticity and BDNF regulation by microRNAs" is overwhelmed with references.

There should be a paragraph for limitations.

The review is too short and could be expanded.

"Some microRNAs can be are exported from cells" should be corrected.

"Together, those findings suggest a mechanistic crosstalk driven by microRNAs 125 between inflammation in peripheral systems and neurodegenerative processes" Neurodegenerative processes is a layman's term.

"Some studies showed that Sonic hedgehog (Shh) is able to relief the suppression ex-194 erted by miR-206 on BDNF synthesis" not clear what this means.

The article should have extended conclusions.

Overall, the English language needs to be checked.

Author Response

Please find the referred revisions attached and changes in the manuscript marked in yellow.

Round 2

Reviewer 2 Report

Comments and Suggestions for Authors

The authors address all of the reviewer's concerns 

Reviewer 3 Report

Comments and Suggestions for Authors I looked at the article carefully and observed that the citations were reduced.
I, therefore, approve the acceptance.